# You Are the Best Reviewer of Your Own Papers: An Owner-Assisted Scoring Mechanism

**Weijie J. Su**
Department of Statistics and Data Science
University of Pennsylvania
`suw@wharton.upenn.edu`

## Abstract

I consider the setting where reviewers offer very noisy scores for a number of items for the selection of high-quality ones (e.g., peer review of large conference proceedings) whereas the owner of these items knows the true underlying scores but prefers not to provide this information. To address this withholding of information, in this paper, I introduce the *Isotonic Mechanism*, a simple and efficient approach to improving on the imprecise raw scores by leveraging certain information that the owner is incentivized to provide. This mechanism takes as input the ranking of the items from best to worst provided by the owner, in addition to the raw scores provided by the reviewers. It reports adjusted scores for the items by solving a convex optimization problem. Under certain conditions, I show that the owner's optimal strategy is to honestly report the true ranking of the items to her best knowledge in order to maximize the expected utility. Moreover, I prove that the adjusted scores provided by this owner-assisted mechanism are indeed significantly more accurate than the raw scores provided by the reviewers. This paper concludes with several extensions of the Isotonic Mechanism and some refinements of the mechanism for practical considerations.

## 1 Introduction

Designing mechanisms for evaluating a set of items by aggregating scores from agents has applications in a wide range of domains [7, 2, 25]. As an example of these applications, reviewers are asked to score submissions for conference proceedings on the basis of certain criteria, and the scores serve as a crucial factor in accepting or rejecting the submissions. Ideally, one wishes to design a mechanism that is expected to produce scores that adequately reflect the quality of the submissions.

In the last few years, however, the explosion in the number of submissions to many artificial intelligence and machine learning conferences has made it very challenging to maintain high-quality scores from peer review. For instance, the number of submissions to NeurIPS—a leading machine learning conference—has grown from 1,673 in 2014 to a record-high of 9,454 in 2020 [18, 13]. On the other hand, the number of qualified reviewers—those, for example, who have published at least one paper in a premier machine learning conference—simply cannot keep up with the growth of the submission size [24]. This significant disparity imposes significant burdens on the qualified reviewers, thereby reducing the average amount of time spent on reviewing a paper and, worse, pushing program chairs to solicit novice reviewers for top-tier machine learning conferences, including NeurIPS, ICML, AAAI, and many others [26].

Given the fundamental role of peer review in scientific research, the consequence of this reality has started to impede the advance of machine learning research [17, 8, 28]. In the famous NIPS experiment in 2014 [18], review scores are observed to have a surprisingly high level of arbitrariness. The situation has presumably become even worse, as the number of submissions in 2020 was

35th Conference on Neural Information Processing Systems (NeurIPS 2021).

more than five times that when the experiment was conducted. While a moderate level of review arbitrariness reflects the diversity of the scientific community, such a high level as in the 2014 NIPS experiment would very likely lead to acceptance of many low-quality papers and, correspondingly, rejection of a high volume of high-potential papers. In the long term, the poor reliability of peer review might erode the public confidence in these machine learning conference proceedings, a challenge that the scientific community cannot afford to ignore [23].

To address this challenge, in this paper I introduce a mechanism that allows one to produce more reliable scores that are modified from the raw review scores. While it seems unrealistic to extend the reviewer pool or assign more papers to each reviewer, it is possible—though rarely done—to extract information from the authors and incorporate it into the improvement of scores (see, e.g., [25]). In fact, an author often submits multiple papers simultaneously to the same conference and, moreover, is likely to be among the most capable persons to assess the quality of her submissions. Unfortunately, if asked, the author has an incentive to report higher scores than what she believes so as to maximize the utility. To reconcile this dilemma, instead, my mechanism requires the author to report a ranking of her submissions in decreasing order of the quality. With this ranking in place, the mechanism solves its adjusted scores from a simple convex programming problem—which is in the form of isotonic regression [4]—using the raw review scores and the ranking as data. Thus, this new mechanism is referred to as the Isotonic Mechanism.

The Isotonic Mechanism comes with several appealing properties that guarantee its superiority over the use of the raw review scores for decision making. First, I show that a rational author's *unique* optimal strategy is to report the true ranking of her submissions under the assumption that the utility of the author is an additive convex function over all the submissions. Even in the case where the author is not completely certain about the ranking of the "true underlying" scores, the author is better off reporting the most accurate ranking as far as she knows. Second, I prove that the Isotonic Mechanism supplied with the optimal strategy taken by the author strictly improves the accuracy of estimating the true underlying scores.

Our further analysis reveals that the gain of the Isotonic Mechanism over the raw scores becomes especially substantial if the variability of the scores provided by reviewers is high and the number of submissions by the author is large—the very situation that many machine learning conferences are faced with. While the high variability often results from the reduction of review time per submission and the arbitrariness of novice reviewers [10], more recently it has become increasingly common for an author to submit multiple papers to one single conference. For example, there are 133 authors who submitted at least five papers to ICLR in 2020, which constitutes a significant portion of the total 2,594 submissions. Even more surprisingly, the maximum number of submissions authored by one single individual is as high as 32 in this conference [27]. These ever-growing statistics render the Isotonic Mechanism an ideal approach to improving peer review in these scenarios.

In relating to the literature, the Isotonic Mechanism is largely orthogonal to many existing approaches to obtaining better review scores [19, 11, 29]. In this line of research, the primary focus is to develop mechanisms that incentivize reviewers to provide more accurate scores or to sharpen the scores by modeling the generation of scores. In stark contrast, the Isotonic Mechanism solely exploits the information from the authors' side in addition to the raw scores. This orthogonality suggests the promise of incorporating the underlying idea of the Isotonic Mechanism into many existing approaches. For completeness, I remark that an emerging line of research in the mechanism design area leverages additional author information to improve peer review [1, 22, 14, 21].

## 2 Methodology

In this section, I introduce the Isotonic Mechanism in detail, and provide theoretical guarantees under certain conditions showing that (1) the owner/author would report her true ranking of her items/submissions in order to maximize the expected utility, and (2) this mechanism improves the overall statistical estimation efficiency for evaluating the papers. Throughout this paper, I consider an owner of the items who is fully rational in that her ultimate goal is to maximize the expectation of the overall utility as much as she can.

Consider $n$ items and let their true scores be denoted by $R_1, R_2, \ldots, R_n$. The reviewers rate the $n$ items as $y_1, \ldots, y_n$, which are given by

$$y_i = R_i + z_i \tag{1}$$

for $i = 1, \ldots, n$. Above, $z_i$'s are the noise terms. In our setup, the owner of the $n$ items is asked to provide a ranking of the items, denoted by $\pi$, which is a permutation of $1, 2, \ldots, n$. Ideally, the permuted true scores

$$\pi \circ \boldsymbol{R} := (R_{\pi(1)}, R_{\pi(2)}, \ldots, R_{\pi(n)})$$

are (approximately) in the nonincreasing order, but it does not necessarily have to be this case, as the owner has complete freedom to report any ranking. In our mechanism, the ranking serves as a shape-restricted constraint that shall facilitate better estimation of the true underlying scores $R_1, \ldots, R_n$ based on the raw scores $\boldsymbol{y} = (y_1, \ldots, y_n)$. More precisely, taking as input the raw scores $\boldsymbol{y}$ by the reviewers and the ranking $\pi$ by the owner, our mechanism outputs scores $\widehat{\boldsymbol{R}} := (\widehat{R}_1, \ldots, \widehat{R}_n)$ that either serve as the adjusted scores or as a reference for the decision-making by, for example, the program chair.

Now I introduce the Isotonic Mechanism for adjusting the raw scores by the reviewers, provided the input data $\boldsymbol{y}$ and ranking $\pi$. Throughout the paper, $\| \cdot \|$ denotes the Euclidean norm.

> The Isotonic Mechanism reports $\widehat{\boldsymbol{R}}(\pi)$, which is the optimal solution of the following convex program:
>
> $$\min_{\boldsymbol{r}} \quad \frac{1}{2}\|\boldsymbol{y} - \boldsymbol{r}\|^2 \tag{2}$$
> $$\text{s.t.} \quad r_{\pi(1)} \geq r_{\pi(2)} \geq \cdots \geq r_{\pi(n)}.$$

When clear from the context, I omit the dependence on $\pi$ when writing the reported ranking $\widehat{\boldsymbol{R}}$.

In words, the Isotonic Mechanism finds the solution by projecting the raw scores onto the feasible region $\{\boldsymbol{r} : r_{\pi(1)} \geq r_{\pi(2)} \geq \cdots \geq r_{\pi(n)}\}$, which constitutes a *convex* cone that is often referred to as the isotonic cone (up to permutation), hence the name Isotonic Mechanism. Needless to say, the mechanism wishes that the owner can report the true ranking, although it is not necessarily the case. If the reported ranking is $\pi^\star$, the true values $\boldsymbol{R}$ is a feasible point of the program (2) and this program is essentially a restricted least-squares estimator. The projection shall presumably improve the estimation accuracy as it greatly reduces the search space for the true scores.

From an algorithmic angle, an appealing feature of this mechanism is that (2) is a convex quadratic programming problem and, therefore, is tractable. Even better, this program admits a certain structure that allows for a very efficient algorithm called the pool adjacent violators algorithm [16, 3, 6]. Interested readers are referred to [5, 12] for other algorithms for solving isotonic regression.

To proceed, several pressing questions concerning the Isotonic Mechanism must be addressed. What is the optimal strategy of the owner and will she report the true ranking out of her own interest? Moreover, does the mechanism, provided that the owner chooses the optimal strategy, really outperform the raw scores of the reviewers in any sense? These questions will be addressed by the following subsections.

## 2.1 Optimality

To find the optimal strategy of the owner, I first recognize that the owner is rational and aims to maximize her expected utility by supplying any ranking $\pi$ of her choice to the Isotonic Mechanism (2). On the surface, it seems unclear which ranking would maximize the expected utility.

I make the following assumptions for the setup for the mechanism. Before introducing the assumptions, it is worth noting that the next section extends our results to a certain relaxed version of these assumptions.

**Assumption 2.1.** *Given (final) scores $\widehat{R}_1, \ldots, \widehat{R}_n$ of $n$ items in the possession of the owner, then the owner's utility takes the form* $\text{Util}(\widehat{\boldsymbol{R}}) = \sum_{i=1}^{n} U(\widehat{R}_i)$, *where $U$ is a nondecreasing convex function.*

In this assumption, the utility function is *separable* in the sense that it is additive over all the $n$ items. Regarding the convexity assumption of the utility function, note that it boils down to requiring that the marginal utility $U'(r)$ (assume differentiability) is a nondecreasing function. At first glance, it is not consistent with the conventional law of diminishing marginal utility [15]. However, a more careful look first shows that score $\widehat{R}_i$ does not measure the quantity of some matter. Moreover, note

that for peer review in conference proceedings, a high score largely determines whether the paper would be presented as a poster, oral talk, or even as a plenary talk, which really makes a significant difference for a paper in terms of impact. In this spirit, the utility increases rapidly as the score increases and therefore a convexity assumption on the utility sheds light on this reality.

In Section 3 I consider an extension of this assumption. In addition, I discuss this assumption in the practical context of peer review in Section 4.

Notably, the utility in many applications can vanish to zero once the score $r$ is below some value. In view of this fact, examples of such utility functions include $U(r) = \max\{0, ar + b\}$ and $U(r) = \max\{0, e^{ar+b} - c\}$, where $a$ and $c$ are positive constants.

**Assumption 2.2.** *The owner has knowledge of the true ranking of her items. That is, the owner knows which permutation $\pi^\star$ that makes $\pi^\star \circ \boldsymbol{R}$ in nonincreasing order.*

*Remark* 2.1. Turning to Assumption 2.2, it is worthwhile mentioning that the owner only needs to compare the true values instead of knowing the exact values of $R_i$'s in order to report the true ranking. In Section 3, I relax this assumption by considering an owner who possesses only partial knowledge of the true ranking.

**Assumption 2.3.** *The noise $(z_1, \ldots, z_n)$ follows an exchangeable distribution in the sense that $(z_1, \ldots, z_n)$ has the same probability distribution in $\mathbb{R}^n$ as $\rho \circ \boldsymbol{z} := (z_{\rho(1)}, \ldots, z_{\rho(n)})$ for any permutation $\rho$ of $1, \ldots, n$.*

*Remark* 2.2. This assumption includes the often-assumed setting where $(z_1, \ldots, z_n)$ is comprised of independent and identically distributed random variables. This generality is useful in the case where the noise terms involve a common latent factor—for example, a recent trend in research directions—which invalidates independence but the exchangeability remains true. Furthermore, it is beneficial to note that if the score of the $i$-th paper is averaged over several reviewers, the noise terms may have different variances. However, our assumption remains valid if I can treat the number of reviewers assigned to each paper as an independent and identically distributed random variable. In addition, technically speaking, our assumptions do not require the noise to have zero mean, though I do not intend to emphasize this generalization.

Interestingly, the following theorem shows that reporting the true ranking is just the optimal strategy. Recall that the true ranking $\pi^\star$ is a permutation of $1, \ldots, n$ such that the permuted true scores obey $R_{\pi^\star(1)} \geq R_{\pi^\star(2)} \geq \cdots \geq R_{\pi^\star(n)}$, and the expected utility

$$\mathbb{E}\operatorname{Util}(\widehat{\boldsymbol{R}}) = \mathbb{E}\left[\sum_{i=1}^{n} U(\widehat{\beta}_i)\right] \tag{3}$$

is taken over the randomness of the noise terms in (1).

**Theorem 1.** *Under Assumptions 2.1-2.3, the expected utility* (3) *is maximized when the Isotonic Mechanism is provided with the true ranking $\pi^\star$. That is, the owner's optimal strategy is to honestly report the true ranking.*

To better appreciate this theorem, note that this optimal strategy is "computationally free" in the sense that reporting the true ranking does not require further effort from the owner under our assumptions. In particular, the optimal strategy does not rely on the raw scores provided by the reviewers. Moreover, as revealed by the proof in Section 2.2, if the true scores $\boldsymbol{R}$ do not contain ties and the function $U$ is strictly convex, the optimality claimed in Theorem 1 is strict in the sense that $\mathbb{E}\operatorname{Util}(\widehat{\boldsymbol{R}}(\pi)) < \mathbb{E}\operatorname{Util}(\widehat{\boldsymbol{R}}(\pi^\star))$ for any ranking $\pi$ that is not identical to $\pi^\star$. The gap between the two utility terms can be arbitrarily large when the true underlying scores are widely different from each other. Even in the presence of ties, the inequality above remains strict unless $\pi$ is also a true ranking, thereby only differing from $\pi^\star$ in the indices corresponding to the ties.

In light of Theorem 1, in the remainder of this section I assume that the owner reports the true ranking out of self-interest. The next step is to study the performance of the Isotonic Mechanism from a social welfare viewpoint. Explicitly, for example, are the adjusted scores more accurate than the raw scores? The following theorem answers this question in the affirmative.

**Theorem 2.** *Under Assumptions 2.1-2.3, the Isotonic Mechanism improves the estimation accuracy of the true underlying scores in the sense that*

$$\mathbb{E}\left[\sum_{i=1}^{n}\left(\widehat{R}_i(\pi^\star) - R_i\right)^2\right] \leq \mathbb{E}\left[\sum_{i=1}^{n}(y_i - R_i)^2\right].$$

*Remark* 2.3. Above, the $\ell_2$ loss is chosen mainly for technical convenience. An interesting question for future research is to analyze the performance using other loss functions.

Together with Theorem 1, Theorem 2 demonstrates the superiority of the Isotonic Mechanism over the use of the raw scores by the reviewers. Furthermore, the following result shows that the gained improvement becomes more significant in the case when the number of items and the noise variance are large.

To state this theorem, below I impose more refined assumptions on the noise terms. Besides, I let

$$V(\boldsymbol{R}) := \sum_{i=1}^{n-1} |R_{\pi^\star(i+1)} - R_{\pi^\star(i)}| = \max_i R_i - \min_i R_i$$

denote the total variation of $\boldsymbol{R}$.

**Theorem 3.** *In additions to Assumptions 2.2 and 2.1, assume that $z_1, \dots, z_n$ are independent and identically distributed normal random variables $\mathcal{N}(0, \sigma^2)$. Letting both $\sigma > 0$ and $V > 0$ be fixed, I have*

$$0.4096 + o_n(1) \le \frac{\sup_{\boldsymbol{R}:V(\boldsymbol{R})\le V} \mathbb{E}\left[\sum_{i=1}^{n}\left(\widehat{R}_i(\pi^\star) - R_i\right)^2\right]}{n^{\frac{1}{3}}\sigma^{\frac{4}{3}}V^{\frac{2}{3}}} \le 7.5625 + o_n(1),$$

*where both $o_n(1) \to 0$ as $n \to \infty$.*

*Remark* 2.4. This theorem is adapted from a well-known result of isotonic regression (see, e.g., Theorem 2.3 in [30]) and, thus the proof is omitted. As an aside, the risk bounds should not be interpreted as being always increasing with the total variation bound $V$. To see this, note that when $R_{\pi^\star(i+1)} \gg R_{\pi^\star(i)}$ for all $i$, then the raw scores $y_1, \dots, y_n$ satisfy the isotonic constraint (2) with $\pi$ set to $\pi^\star$ with high probability. As a consequence, the risk of the Isotonic Mechanism is roughly equal to that of the raw scores. Even if the total variation $V$ tends to infinity, the risk stays constant.

Loosely speaking, this theorem says that the risk of the Isotonic Mechanism is of order $O(n^{1/3}\sigma^{4/3})$. For comparison, note that the risk of the raw scores is $\mathbb{E}\left[\sum_{i=1}^{n}(y_i - R_i)^2\right] = \mathbb{E}\left[\sum_{i=1}^{n} z_i^2\right] = n\sigma^2$. Recognizing their dependence on the noise level $\sigma$, I see that the Isotonic Mechanism is especially preferable when $\sigma$ is large or, put differently, when the reviewers give very noisy scores. Moreover, the Isotonic Mechanism has a risk that grows much slower with respect to the number of items $n$.

## 2.2 Sketch of Proof

This section is devoted to proving Theorem 1 and we defer the proof of Theorem 2 to the Appendix. We need the following two lemmas for the proof of Theorem 1.

**Lemma 2.4.** *Let $\boldsymbol{x} = (x_1, \dots, x_n) \succeq \boldsymbol{y} = (y_1, \dots, y_n)$ in the sense that $x_1 \ge y_1, x_1 + x_2 \ge y_1 + y_2, \dots, x_1 + \dots + x_{n-1} \ge y_1 + \dots + y_{n-1}$ and $x_1 + \dots + x_n = y_1 + \dots + y_n$. Let $\boldsymbol{x}^+$ and $\boldsymbol{y}^+$ be the projections of $\boldsymbol{x}$ and $\boldsymbol{y}$ onto the isotonic cone $\{\boldsymbol{r} : r_1 \ge r_2 \ge \dots \ge r_n\}$, respectively. Then, we have $\boldsymbol{x}^+ \succeq \boldsymbol{y}^+$.*

**Lemma 2.5** (Hardy–Littlewood–Pólya inequality)**.** *Let $f$ be a convex function. Assume that $\boldsymbol{x}$ and $\boldsymbol{y}$ are nonincreasing vectors (that is, $x_1 \ge \dots \ge x_n$ and $y_1 \ge \dots \ge y_n$) and $\boldsymbol{x} \succeq \boldsymbol{y}$. Then, we have $\sum_{i=1}^{n} f(x_i) \ge \sum_{i=1}^{n} f(y_i)$.*

Lemma 2.5 is a well-known result in the theory of majorization. See its proof in [20].

In contrast, Lemma 2.4 is new to the literature and its proof requires some novel elements. This is partly because the definition of majorization in this lemma is slightly different from that in the literature [20]. In the literature, $\boldsymbol{x} \succeq \boldsymbol{y}$ if the condition in Lemma 2.4 holds after ordering both $\boldsymbol{x}, \boldsymbol{y}$ from the largest to the smallest. To this end, we need the following definition and Lemma 2.7.

**Definition 2.6.** We call $\boldsymbol{z}^1$ an upward swap of $\boldsymbol{z}^2$ if there exists $1 \le i < j \le n$ such that $z_k^1 = z_k^2$ for all $k \ne i, j$ and $z_i^1 + z_j^1 = z_i^2 + z_j^2, z_i^1 \ge z_i^2$.

This definition amounts to saying that $\boldsymbol{z}^1$ is an upward swap of $\boldsymbol{z}^2$ if the former can be derived by transporting some mass from the entry of $\boldsymbol{z}^2$ to an earlier entry. It is easy to check that $\boldsymbol{z}^1 \succeq \boldsymbol{z}^2$ if $\boldsymbol{z}^1$ is an upward swap of $\boldsymbol{z}^2$.

**Lemma 2.7.** *Let $\boldsymbol{x} \succeq \boldsymbol{y}$. Then there exists an integer $m$ and $\boldsymbol{z}^1, \ldots, \boldsymbol{z}^m$ such that $\boldsymbol{z}^1 = \boldsymbol{x}, \boldsymbol{z}^m = \boldsymbol{y}$, and $\boldsymbol{z}^l$ is an upward swap of $\boldsymbol{z}^{l+1}$ for $l = 1, \ldots, m-1$.*

Due to space constraints, we relegate the proofs of Lemma 2.7 and Lemma 2.4 to the Appendix.

*Proof of Theorem 1.* Without loss of generality, assume that $R_1 \geq R_2 \geq \cdots \geq R_n$. In this case, the Isotonic Mechanism is

$$\min \quad \frac{1}{2}\|\boldsymbol{y} - \boldsymbol{r}\|^2$$
$$\text{s.t. } r_1 \geq r_2 \geq \cdots \geq r_n,$$

where $\boldsymbol{y} = \boldsymbol{R} + \boldsymbol{z}$. As is clear, the solution, denoted $\widehat{\boldsymbol{R}}$, is obtained by projecting $\boldsymbol{y}$ onto the isotonic cone $C = \{\boldsymbol{r} : r_1 \geq r_2 \geq \cdots \geq r_n\}$. Now consider that the owner proposes a different ranking, $\pi$, which corresponds to

$$\min \quad \frac{1}{2}\|\boldsymbol{y} - \boldsymbol{r}\|^2$$
$$\text{s.t.} \quad r_{\pi(1)} \geq r_{\pi(2)} \geq \cdots \geq r_{\pi(n)},$$

which is equivalent to

$$\min \quad \frac{1}{2}\|\pi \circ \boldsymbol{y} - \boldsymbol{r}\|^2$$
$$\text{s.t.} \quad r_1 \geq r_2 \geq \cdots \geq r_n$$

after an appropriate permutation operation. In this case, the owner's ratings be can obtained by projecting $\boldsymbol{y}_\pi$ onto the isotonic cone, and then pulling back the original permutation by applying $\pi^{-1}$. Now, defining $\pi \circ \boldsymbol{a}$ as $\pi \circ \boldsymbol{a} = (a_{\pi(1)}, a_{\pi(2)}, \ldots, a_{\pi(n)})$ for a vector $\boldsymbol{a}$, we observe that $\pi \circ \boldsymbol{y} = \pi \circ \boldsymbol{R} + \pi \circ \boldsymbol{z}$. For simplicity, we let $\text{Proj}_C$ denote the projection operator. Consider the two projections associated with the two rankings, namely $\text{Proj}_C(\boldsymbol{y})$ *and* $\pi^{-1} \circ \text{Proj}_C(\pi \circ \boldsymbol{y})$. Due to our construction of the utility function, the owner is indifferent between $\pi^{-1} \circ \text{Proj}_C(\pi \circ \boldsymbol{y})$ and $\text{Proj}_C(\pi \circ \boldsymbol{y})$. Next, making use of our assumption on the noise distribution, we have $\pi \circ \boldsymbol{R} + \pi \circ \boldsymbol{z} \stackrel{d}{=} \pi \circ \boldsymbol{R} + \boldsymbol{z}$. Now we compare $\text{Proj}_C(\boldsymbol{R} + \boldsymbol{z})$ *and* $\text{Proj}_C(\pi \circ \boldsymbol{R} + \boldsymbol{z})$ in terms of majorization. Using the simple but important fact that $\boldsymbol{R} + \boldsymbol{z} \succeq \pi \circ \boldsymbol{R} + \boldsymbol{z}$, it follows from Lemma 2.4 that $\text{Proj}_C(\boldsymbol{R} + \boldsymbol{z}) \succeq \text{Proj}_C(\pi \circ \boldsymbol{R} + \boldsymbol{z})$.

Therefore, Lemma 2.5 gives

$$\sum_{i=1}^n U\left(\text{Proj}_C(\boldsymbol{R} + \boldsymbol{z})_i\right) \geq \sum_{i=1}^n U\left(\text{Proj}_C(\pi \circ \boldsymbol{R} + \boldsymbol{z})_i\right),$$

from which we readily get

$$\mathbb{E}\left[\sum_{i=1}^n U\left(\text{Proj}_C(\boldsymbol{R} + \boldsymbol{z})_i\right)\right] \geq \mathbb{E}\left[\sum_{i=1}^n U\left(\text{Proj}_C(\pi \circ \boldsymbol{R} + \boldsymbol{z})_i\right)\right].$$

That is, we have $\text{Util}_{\text{true ranking}} \geq \text{Util}_\pi$ for an arbitrary ranking $\pi$. This finishes the proof.

$\square$

## 3 Extensions

In this section, we extend our main results by relaxing the assumptions made in Section 2.1. Our primary aim is to show that the rational owner would continue to report the true ranking of her items in more general settings. Omitted proofs are deferred to the Appendix.

**Ranking in a block form.** Suppose that the owner only knows partial information of the true ranking in a block sense: let $n_1, \ldots, n_m$ be positive integers satisfying $n_1 + \cdots + n_m = n$ and $\{1, 2, \ldots, n\}$ be partitioned into $\{1, 2, \ldots, n\} = I_1^\star \cup I_2^\star \cup \cdots \cup I_m^\star$ such that $|I_i| = n_i$ for $i = 1, \ldots, m$. The owner knows the true ranking satisfies

$$\boldsymbol{R}_{I_1^\star} \geq \boldsymbol{R}_{I_2^\star} \geq \cdots \geq \boldsymbol{R}_{I_m^\star}, \tag{4}$$

but the ranking within each block is completely unknown to the owner. The owner is asked to report an (ordered) partition $I_1, \ldots, I_m$ of $\{1, \ldots, n\}$ such that $|I_i| = n_i$ for $i = 1, \ldots, m$. Now we consider the following $(n_1, \ldots, n_m)$-block Isotonic Mechanism:

$$\min_{\boldsymbol{r}} \quad \frac{1}{2}\|\boldsymbol{y} - \boldsymbol{r}\|^2 \tag{5}$$
$$\text{s.t.} \quad \boldsymbol{r}_{I_1} \geq \boldsymbol{r}_{I_2} \geq \cdots \geq \boldsymbol{r}_{I_m}.$$

Note that it is also a convex optimization program.

**Assumption 3.1.** *The owner has knowledge of the true $(n_1, \ldots, n_m)$-block ranking of her items. That is, the owner knows which $(n_1, \ldots, n_m)$-sized partition that satisfies* (4).

**Theorem 4.** *Under Assumptions 2.1, 3.1, and 2.3, the expected utility* (3) *is maximized when the block Isotonic Mechanism is provided with the true $(n_1, \ldots, n_m)$-sized (ordered) partition.*

*Remark* 3.1. The theorem above can intuitively be extended in a hierarchical manner. That is, for each block $I_i$, the owner can further provide a ranking within this block either in a partial sense or complete sense. The theorem shall continue to be valid and a rigorous treatment of this extension is left for future research.

Denote by $\boldsymbol{I} := (I_1, \ldots, I_m)$. Let $\pi_{\boldsymbol{I},\boldsymbol{y}}$ be formed by padding the $m$ blocks while setting the entries of $\boldsymbol{y}$ within each block in nonincreasing order. That is, the set $\{\pi_{\boldsymbol{I},\boldsymbol{y}}(1), \ldots, \pi_{\boldsymbol{I},\boldsymbol{y}}(n_1)\} = I_1$ and $\boldsymbol{y}_{\pi_{\boldsymbol{I},\boldsymbol{y}}(1)} \geq \boldsymbol{y}_{\pi_{\boldsymbol{I},\boldsymbol{y}}(2)} \geq \cdots \geq \boldsymbol{y}_{\pi_{\boldsymbol{I},\boldsymbol{y}}(n_1)}$, and likewise for the following $m - 1$ blocks. For example, let $n = 4, I_1 = \{1, 3\}, I_2 = \{2, 4\}$, and $\boldsymbol{y} = (4.4, 6.6, 5, -1)$, then $(\pi_{\boldsymbol{I},\boldsymbol{y}}(1), \pi_{\boldsymbol{I},\boldsymbol{y}}(2), \pi_{\boldsymbol{I},\boldsymbol{y}}(3), \pi_{\boldsymbol{I},\boldsymbol{y}}(4)) = (3, 1, 2, 4)$ and $\boldsymbol{y}_{\pi_{\boldsymbol{I},\boldsymbol{y}}} = (5, 4.4, 6.6, -1)$.

**Robustness to inconsistencies.** Next, we extend Theorem 1 to the setting where the owner might give a ranking that is not consistent with the true values. Imagine that the owner is faced with the problem of choosing between two rankings $\pi_1, \pi_2$. While $\pi_1$ nor $\pi_2$ might not render the true values $\boldsymbol{R}$ in nonincreasing order, we assume that the former is more faithful with respect to $\boldsymbol{R}$ than the latter in the sense that

$$\pi_1 \circ \boldsymbol{R} \succeq \pi_2 \circ \boldsymbol{R}. \tag{6}$$

To better understand when this condition holds, we say that $\pi_1$ is an upward shuffle of $\pi_2$ if there exists two indices $1 \leq i < j \leq n$ such that

$$R_{\pi_1(i)} = R_{\pi_2(j)} > R_{\pi_1(j)} = R_{\pi_2(i)}$$

and $\pi_1(k) = \pi_2(k)$ for all $k \neq i, j$. As is clear, $(\pi_1, \pi_2)$ satisfies (6). In general, (6) is satisfied if $\pi_1$ can be derived by sequentially upwardly shuffling from $\pi_2$, by recognizing the transitivity of majorization. However, it is worth noting that it is *not* correct that any pair $(\pi_1, \pi_2)$ satisfying (6) can always be sequentially shuffled from one to another. An example is $\pi_1 \circ \boldsymbol{R} = (100, 0, 1, 10)$ and $\pi_2 \circ \boldsymbol{R} = (10, 1, 0, 100)$.

The following result demonstrates that Theorem 1 extends to this fault-tolerant setting.

**Theorem 5.** *Under Assumptions 2.1 and 2.3, the owner in the face of choosing between $\pi_1$ and $\pi_2$ satisfying* (6), *which is known to the owner, would be better off reporting $\pi_1$ rather than $\pi_2$ in terms of the overall utility.*

**Non-separable utility functions.** A function $f : \mathbb{R}^n \to \mathbb{R}$ is called a Schur-convex function if it is symmetric in its $n$ coordinates and $f(\boldsymbol{x}) \geq f(\boldsymbol{y})$ for any $\boldsymbol{x}, \boldsymbol{y}$ such that $x_1 \geq \cdots \geq x_n, y_1 \geq \cdots \geq y_n$ and $\boldsymbol{x} \succeq \boldsymbol{y}$ as defined in Lemma 2.4. As shown by the proofs in Section 2.2, the utility $\text{Util}(\boldsymbol{r})$ given in Assumption 2.1 is Schur-convex. More generally, when $f$ is differentiable and symmetric, then $f$ is Schur-convex if and only if

$$(r_i - r_j) \left( \frac{\partial f(\boldsymbol{r})}{\partial r_i} - \frac{\partial f(\boldsymbol{r})}{\partial r_j} \right) \geq 0$$

for all $\boldsymbol{r} \equiv (r_1, \ldots, r_n)$ (see, e.g., [9]).

Recognizing that the final step in the proof of Theorem 1 relies on the majorization property, we readily obtain the following result.

**Proposition 3.2.** *In addition to Assumptions 2.2 and 2.3, assume that the function $\text{Util}(\boldsymbol{r})$ is Schur-convex. Then, the owner's optimal strategy is to honestly report the true ranking of her items.*

**True score-dependent utility.** Intuitively, the utility of each items shall depend on the true score of the item. An owner may value more about the acceptance of a high-quality item. In this spirit, we consider utility function that takes the following form:

$$\text{Util}(\widehat{\boldsymbol{R}}) = \sum_{i=1}^{n} U(\widehat{R}_i; R_i), \tag{7}$$

where we assume that $U(r; R)$ is convex is in its first argument and satisfies

$$\frac{\partial U(r; R)}{\partial r} \geq \frac{\partial U(r; R')}{\partial r}$$

whenever $R > R'$. For example, consider $U(r; R) = g(r)h(R)$ where $g \geq 0$ is convex and $h \geq 0$ is a nondecreasing differential function.

The following result shows that Theorem 1 remains valid under this generalization.

**Proposition 3.3.** *In addition to Assumptions 2.2 and 2.3, assume that the function* $\text{Util}(\boldsymbol{r})$ *is given in* (7). *Then, the owner's optimal strategy is to honestly report the true ranking of her items.*

*Proof of Proposition 3.3.* The proof of this proposition is basically the same as that of Theorem 1 except for a minor modification. Explicitly, observe that

$$\sum_{i=1}^{n} U(\widehat{R}_i; R_i)$$

is always upper bounded by

$$\sum_{i=1}^{n} U(\widehat{R}_i; R_{\rho(i)})$$

for a permutation $\rho$ such that $\widehat{R}_i$'s and $R_{\rho(i)}$'s have the same ranking in decreasing order. An important fact is that in reporting the true ranking, the utility has already been maximized over all permutations. The remaining part of the proof is the same as that of Theorem 1.

$\square$

**Soft constraints.** Conceivably, one would argue that the hard constraints provided by the isotonic constraint seems too strong, especially in the case where the owner might not be completely sure about the ranking of her items. In this case, a simple adjustment is to consider the following convex combination as a candidate of scores:

$$\widehat{\boldsymbol{R}}' := \theta \widehat{\boldsymbol{R}}(\pi^\star) + (1 - \theta)\boldsymbol{y},$$

where $0 < \theta < 1$. Owing to the convexity of the risk, Theorem 2 remains correct.

**Proposition 3.4.** *Under the assumptions of Theorem 2, we have*

$$\mathbb{E}\left[\sum_{i=1}^{n}\left(\widehat{R}_i' - R_i\right)^2\right] \leq \mathbb{E}\left[\sum_{i=1}^{n}(y_i - R_i)^2\right].$$

Note that this generalization does not change the mechanism in that the owner still reports the ranking in view of the isotonic regression. More generally, we consider the following optimization program:

$$\min_{\boldsymbol{r}} \frac{1}{2}\|\boldsymbol{y} - \boldsymbol{r}\|^2 + \text{Pen}(\boldsymbol{r}),$$

where the (nonnegative) penalty term $\text{Pen}(\boldsymbol{r})$ encourages isotonic monotonicity in the sense that, for example, $\text{Pen}(\boldsymbol{r}) = 0$ if $\boldsymbol{r}$ is in the isotonic cone $C$ and otherwise $\text{Pen}(\boldsymbol{r}) > 0$. A choice of this penalty is

$$\text{Pen}(\boldsymbol{r}) = \lambda \sum_{i=1}^{n-1} (r_{\pi(i+1)} - r_{\pi(i)})_+,$$

where $\lambda > 0$, $\pi$ is the ranking reported by the owner and $x_+ := \max\{x, 0\}$.

As $\lambda \to \infty$, the penalty of the violation of the isotonic constraint tends to infinity. Therefore, we have the following simple fact.

**Proposition 3.5.** *The Isotonic Mechanism can be (asymptotically) recovered by setting $\lambda \to \infty$ in (8).*

However, the coupling strategy is invalid when it comes to proving for this new mechanism. While can we prove that Theorem 1 holds for this $\lambda$-dependent mechanism when $n = 2$, we leave the proof or disproof of the general case $n \geq 3$ for future research.

**Proposition 3.6.** *When $n = 2$, the mechanism above is truth-optimal.*

## 4  Discussion

In this paper, I have introduced the Isotonic Mechanism to incentivize the author of a set of papers to report the true ranking, thereby obtaining more accurate scores for decision making. The mechanism is conceptually simple and computationally efficient. Owing to the appealing optimality guarantees of this mechanism, it is likely that the use of this mechanism can to some extent alleviate the poor quality of review scores in many large machine learning conferences. Toward the employment of this mechanism, an intermediate step is to test the use of this mechanism. For example, conference organizers can require that a randomly selected group of authors submit the ranking and use the outcome of the Isotonic Mechanism for this selected group as a test.

In closing, I propose several directions to tackle practical challenges in applying the Isotonic Mechanism. First, the assumption of convex utility, which is critical for the optimality of the mechanism, might not always hold for some authors. As argued in Section 2.1, the utility is convex or approximately convex if the author aims for higher scores in order to be considered for more visibility and paper awards, beyond the acceptance of her papers. In contrast, if nothing more than acceptance or rejection makes a difference to the author, one may turn to a thresholded utility function of the form

$$U(r) = \begin{cases} 0, & \text{if } r < r_0, \\ u, & \text{if } r \geq r_0, \end{cases}$$

where $u$ and $r_0$ are some constants. An important question for future work is, therefore, to extend the Isotonic Mechanism in the face of non-convex utility.

From a practical standpoint, another direction is to incorporate strategic behaviors of both sides of the author and reviewers. The author might deliberately submit additional very low-quality papers and rank these as the worst. In this case, the Isotonic Mechanism tends to push upward the scores for "normal" papers, thereby increasing the chance of acceptance for these papers which the author cares about. A possible approach to overcoming this issue is to set a screening step to filter very low-quality submissions. Moreover, the Isotonic Mechanism enables the reviewers to have an influence on the author's other papers that are examined by a different set of reviewers. It is interesting to investigate how the reviewers would act for self-interest and improve on the mechanism to confront any violation of ethical codes.

Third, while Theorem 3 shows the great benefits of using the Isotonic Mechanism, it would be of interest to perform experiments to quantify the magnitude of the benefits. As of finishing this paper, the NeurIPS 2021 Program Chairs asked each author to "rank their papers in terms of their own perception of the papers' scientific contributions to the NeurIPS community," though the data was not used for making decisions. Nevertheless, an interesting empirical study is to investigate what would be the outcome if the Isotonic Mechanism were used for NeurIPS 2021.

Last, a critical advance is to extend the Isotonic Mechanism to the setting where each item/paper is owned/authored by multiple owners/authors. This is a necessary step toward applying this mechanism to large machine learning conferences, where most papers are written by multiple authors. Simply invoking the Isotonic Mechanism for each author does not seem to be a good solution. For example, an author can adversarially rank a good paper co-authored by, say Bob, as the lowest so as to put her papers that are not co-authored by Bob in an advantageous position. From a different angle, would the use of the Isotonic Mechanism in an appropriate form discourage guest authorship? I leave this challenging extension for future work.

**Limitations and societal impacts.** A clear limitation of our work lies in its assumptions which our theoretical results rely on. Given the pivotal role of peer review in conference proceedings and the growing impact of AI on our society, we believe that the Isotonic Mechanism would have substantial societal impacts, provided that it would be used someday.

## Acknowledgments and Disclosure of Funding

I am grateful to Nihar Shah and Haifeng Xu for very helpful and enlightening discussions. I would also like to thank the anonymous reviewers for their constructive comments that helped improve the presentation of this work. This work was supported in part by NSF through CAREER DMS-1847415 and CCF-1934876, an Alfred Sloan Research Fellowship, and the Wharton Dean's Research Fund.

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
