# Supplement to "You Are the Best Reviewer of Your Own Papers: An Owner-Assisted Scoring Mechanism"

**Weijie J. Su**
Department of Statistics and Data Science
University of Pennsylvania
suw@wharton.upenn.edu

## 1 Proofs for Section 2

This section is devoted to proving Theorems 1 and 2. We need the following two lemmas for the proof of Theorem 1.

Now, we turn to the proof of Theorem 2. The following proof of this theorem follows from some basic properties of convex sets.

*Proof of Theorem 2.* Consider the (possibly degenerate) triangle formed by $\boldsymbol{y}, \boldsymbol{R}, \widehat{\boldsymbol{R}}$. We claim that the angle $\angle(\boldsymbol{y}, \widehat{\boldsymbol{R}}, \boldsymbol{R}) \geq 90°$. Taking this claim as given for the moment, we immediately conclude that

$$\|\boldsymbol{y} - \boldsymbol{R}\| = \sqrt{\sum_{i=1}^{n} (y_i - R_i)^2} \geq \|\widehat{\boldsymbol{R}} - \boldsymbol{R}\| = \sqrt{\sum_{i=1}^{n} \left(\widehat{R}_i - R_i\right)^2},$$

as desired.

To finish the proof, suppose on the contrary that $\angle(\boldsymbol{y}, \widehat{\boldsymbol{R}}, \boldsymbol{R}) < 90°$. Then there must exist a point $\boldsymbol{R}'$ on the segment between $\widehat{\boldsymbol{R}}$ and $\boldsymbol{R}$ such that $\|\boldsymbol{y} - \boldsymbol{R}'\| < \|\boldsymbol{y} - \widehat{\boldsymbol{R}}\|$. Since both $\widehat{\boldsymbol{R}}$ and $\boldsymbol{R}$ belong to the (convex) isotonic cone $\{\boldsymbol{r} : r_{\pi^\star(1)} \geq \cdots \geq r_{\pi^\star(n)}\}$, the point $\boldsymbol{R}'$ is also in the isotonic cone. However, this contradicts the fact that $\widehat{\boldsymbol{R}}$ is the unique point of the isotonic cone with the minimal distance to $\boldsymbol{y}$. □

Next, we turn to the proof of Theorem 1.

**Lemma 1.1.** *Let $\boldsymbol{x} = (x_1, \ldots, x_n) \succeq \boldsymbol{y} = (y_1, \ldots, y_n)$ in the sense that $x_1 \geq y_1, x_1 + x_2 \geq y_1 + y_2, \ldots, x_1 + \cdots + x_{n-1} \geq y_1 + \cdots + y_{n-1}$ and $x_1 + \cdots + x_n = y_1 + \cdots + y_n$. Let $\boldsymbol{x}^+$ and $\boldsymbol{y}^+$ be the projections of $\boldsymbol{x}$ and $\boldsymbol{y}$ onto the isotonic cone $\{\boldsymbol{r} : r_1 \geq r_2 \geq \cdots \geq r_n\}$, respectively. Then, we have $\boldsymbol{x}^+ \succeq \boldsymbol{y}^+$.*

**Lemma 1.2** (Hardy–Littlewood–Pólya inequality). *Let $f$ be a convex function. Assume that $\boldsymbol{x}$ and $\boldsymbol{y}$ are nonincreasing vectors (that is, $x_1 \geq \cdots \geq x_n$ and $y_1 \geq \cdots \geq y_n$) and $\boldsymbol{x} \succeq \boldsymbol{y}$. Then, we have*

$$\sum_{i=1}^{n} f(x_i) \geq \sum_{i=1}^{n} f(y_i)$$

This is well-known result in theory of majorization. For a proof of Lemma 1.2, see [1].

35th Conference on Neural Information Processing Systems (NeurIPS 2021), Sydney, Australia.

## 1.1 Proof of Lemma 1.1

*Remark* 1.1. See [2] for a proof of this lemma. Throughout this paper, we say $\boldsymbol{x}$ majorizes $\boldsymbol{y}$ if $\boldsymbol{x} \succeq \boldsymbol{y}$. This definition of majorization is slightly different from that in the literature [1]. In the literature, $\boldsymbol{x} \succeq \boldsymbol{y}$ if the condition in Lemma 1.1 holds after ordering both $\boldsymbol{x}, \boldsymbol{y}$ from the largest to the smallest.

**Definition 1.3.** We call $\boldsymbol{z}^1$ an upward swap of $\boldsymbol{z}^2$ if there exists $1 \le i < j \le n$ such that $z_k^1 = z_k^2$ for all $k \ne i, j$ and $z_i^1 + z_j^1 = z_i^2 + z_j^2, z_i^1 \ge z_i^2$.

This definition amounts to saying that $\boldsymbol{z}^1$ is a upward swap of $\boldsymbol{z}^2$ if the former can be derived by transporting some mass from the an entry of $\boldsymbol{z}^2$ to an earlier entry. It is easy to check that $\boldsymbol{z}^1 \succeq \boldsymbol{z}^2$ if $\boldsymbol{z}^1$ an upward swap of $\boldsymbol{z}^2$.

**Lemma 1.4.** *Let $\boldsymbol{x} \succeq \boldsymbol{y}$. Then there exists an integer $m$ and $\boldsymbol{z}^1, \ldots, \boldsymbol{z}^m$ such that $\boldsymbol{z}^1 = \boldsymbol{x}, \boldsymbol{z}^m = \boldsymbol{y}$, and $\boldsymbol{z}^l$ is an upward swap of $\boldsymbol{z}^{l+1}$ for $l = 1, \ldots, m - 1$.*

*Proof of Lemma 1.4.* We prove by induction. The base case $n = 1$ is clearly true. Suppose this lemma is true for $n$.

Now we aim to prove the lemma for the case $n + 1$. Let $\boldsymbol{z}^1 = \boldsymbol{x} = (x_1, x_2, \ldots, x_{n+1})$ and $\boldsymbol{z}^2 := (y_1, x_1 + x_2 - y_1, x_3, x_4, \ldots, x_{n+1})$. As is clear, $\boldsymbol{z}^1$ is an upward swamp of $\boldsymbol{z}^2$.

Now we consider operations on the components except for the first one. Let $\boldsymbol{x}' := (x_1 + x_2 - y_1, x_3, x_4, \ldots, x_{n+1})$ and $\boldsymbol{y}' := (y_2, \ldots, y_{n+1})$ be derived by removing the first component of $\boldsymbol{z}^2$ and $\boldsymbol{y}$, respectively. These two vectors obey $\boldsymbol{x}' \succeq \boldsymbol{y}'$. To see this, note that $x_1' = x_1 + x_2 - y_1 \ge y_1 + y_2 - y_1 = y_2 = y_1'$, and

$$x_1' + \cdots + x_k' = (x_1 + x_2 - y_1) + x_3 + \cdots + x_{k+1} = \sum_{i=1}^{k+1} x_i - y_1 \ge \sum_{i=1}^{k+1} y_i - y_1 = \sum_{i=2}^{k+1} y_i = y_1' + \cdots + y_k'$$

for $2 \le k \le n - 1$ and

$$x_1' + \cdots + x_n' = (x_1 + x_2 - y_1) + x_3 + \cdots + x_{n+1} = \sum_{i=1}^{n+1} x_i - y_1 = \sum_{i=1}^{n+1} y_i - y_1 = y_1' + \cdots + y_n'.$$

Thus, by induction, there must exist $\boldsymbol{z}'^1, \ldots, \boldsymbol{z}'^m$ such that $\boldsymbol{z}'^1 = \boldsymbol{x}', \boldsymbol{z}'^m = \boldsymbol{y}'$, and $\boldsymbol{z}'^l$ is an upward swap of $\boldsymbol{z}'^{l+1}$ for $l = 1, \ldots, m - 1$. We finish the proof for $n + 1$ by recognizing that $\boldsymbol{z}^1 \equiv \boldsymbol{x}, (y_1, \boldsymbol{z}'^1), (y_1, \boldsymbol{z}'^2), \ldots, (y_1, \boldsymbol{z}'^m) \equiv \boldsymbol{y}$ satisfy the requirement of this lemma.

$\square$

Next, consider the following lemma.

**Lemma 1.5.** *Let $\boldsymbol{x}$ be an upward swap of $\boldsymbol{y}$. Let $\boldsymbol{x}^+$ and $\boldsymbol{y}^+$ be the projections of $\boldsymbol{x}$ and $\boldsymbol{y}$ onto the isotonic cone $\{\boldsymbol{r} : r_1 \ge r_2 \ge \cdots \ge r_n\}$, respectively. Then, we have $\boldsymbol{x}^+ \succeq \boldsymbol{y}^+$.*

The proof of Lemma 1.1 readily follows from the use of Lemmas 1.4 and 1.5. To see this point, for any $\boldsymbol{x}, \boldsymbol{y}$ satisfying $\boldsymbol{x} \succeq \boldsymbol{y}$, note that from Lemma 1.4 we can find $\boldsymbol{z}^1 = \boldsymbol{x}, \boldsymbol{z}^2, \ldots, \boldsymbol{z}^{m-1}, \boldsymbol{z}^m = \boldsymbol{y}$ such that $\boldsymbol{z}^l$ is an upward swap of $\boldsymbol{z}^{l+1}$ for $l = 1, \ldots, m - 1$. Then, Lemma 1.5 asserts that $(\boldsymbol{z}^l)^+ \succeq (\boldsymbol{z}^{l+1})^+$ for all $l = 1, \ldots, m - 1$. Due to the transitivity of majorization, we conclude that $\boldsymbol{x} \succeq \boldsymbol{y}$, thereby proving Lemma 1.1.

The following two lemmas will be used in the proof of Lemma 1.5. We relegate the proofs of these two lemmas to the appendix.

**Lemma 1.6.** *For any $\delta > 0$ and $i = 1, \ldots, n$, we have $(\boldsymbol{x} + \delta \boldsymbol{e}_i)^+ \ge \boldsymbol{x}^+$ in the component-wise sense.*

*Remark* 1.2. Likewise, the proof of Lemma 1.6 reveals that $(\boldsymbol{x} - \delta \boldsymbol{e}_i)^+ \le \boldsymbol{x}^+$. As an aside, recognizing a basic property of isotonic regression that $x_1^+ + \cdots + x_n^+ = x_1 + \cdots + x_n$, we have $\mathbf{1}^\top (\boldsymbol{x} + \delta \boldsymbol{e}_i)^+ = \mathbf{1}^\top \boldsymbol{x}^+ + \delta$, where $\mathbf{1} \in \mathbb{R}^n$ denotes the ones vector.

**Lemma 1.7.** *Denote by $\bar{x}$ the sample mean of $\boldsymbol{x}$. Then $\boldsymbol{x}^+$ has constant entries—that is, $x_1^+ = \cdots = x_n^+$—if and only if*

$$\frac{x_1 + \cdots + x_k}{k} \le \bar{x}$$

*for all $k = 1, \ldots, n$.*

*Proof of Lemma 1.5.* Let $1 \le i < j \le n$ be the indices such that $\boldsymbol{x}_i + \boldsymbol{x}_j = \boldsymbol{y}_i + \boldsymbol{y}_j$ and $\boldsymbol{x}_i \ge \boldsymbol{y}_i$. Write $\delta := \boldsymbol{x}_i - \boldsymbol{y}_i \ge 0$. Then, $\boldsymbol{y} = \boldsymbol{x} - \delta \boldsymbol{e}_i + \delta \boldsymbol{e}_j$, where $\boldsymbol{e}_i, \boldsymbol{e}_j$ are the canonical-basis vectors. If $\delta = 0$, then $\boldsymbol{x}^+ = \boldsymbol{y}^+$ because $\boldsymbol{x} = \boldsymbol{y}$, in which case the lemma holds trivially. In the remainder of the proof, we focus on the nontrivial case $\delta > 0$.

The lemma amounts to saying that $\boldsymbol{x}^+ \succeq (\boldsymbol{x} - \delta \boldsymbol{e}_i + \delta \boldsymbol{e}_j)^+$ for all $\delta > 0$. Due to the continuity of the projection, it is sufficient to prove the following statement: there exists $\delta_0 > 0$ (depending on $\boldsymbol{x}$) such that $\boldsymbol{x}^+ \succeq (\boldsymbol{x} - \delta \boldsymbol{e}_i + \delta \boldsymbol{e}_j)^+$.

Let $I$ be the set of indices where the entries of $\boldsymbol{x}^+$ has the same value as $i$:

$$I = \{k : x_k^+ = x_i^+\}.$$

Similarly, define

$$J = \{k : x_k^+ = x_j^+\}.$$

There are exactly two cases, namely $I = J$ and $I \cap J = \emptyset$.

**Case 1.** Consider the case $I = J$. For convenience, write $I = \{a, a+1, \ldots, b-1, b\}$. By Lemma 1.7, we have

$$\frac{x_a + x_{a+1} + \ldots + x_{a+l-1}}{l} \le \bar{x}_I := \frac{x_a + x_{a+1} + \ldots + x_b}{b - a + 1}$$

for $l = 1, \ldots, b - a + 1$.

Now we consider $\boldsymbol{y} = \boldsymbol{x} - \delta \boldsymbol{e}_i + \delta \boldsymbol{e}_j$ restricted to $I$. Assume that $\delta$ is sufficiently small so that the constant pieces of $\boldsymbol{y}^+$ before and after $I$ are the same as those of $\boldsymbol{x}^+$. Since $a \le i < j \le b$, we have

$$y_a + y_{a+1} + \ldots + y_b = x_a + x_{a+1} + \ldots + x_b.$$

On the other hand, we have

$$y_a + y_{a+1} + \ldots + y_{a+l-1} \le x_a + x_{a+1} + \ldots + x_{a+l-1}$$

since the index $i$ comes earlier than $j$. Taken together, these observations give

$$\frac{y_a + y_{a+1} + \ldots + y_{a+l-1}}{l} \le \frac{y_a + y_{a+1} + \ldots + y_b}{b - a + 1}$$

for all $l = 1, \ldots, b - a + 1$. From Lemma 1.7, it follows that the projection $\boldsymbol{y}^+ = (\boldsymbol{x} - \delta \boldsymbol{e}_i + \delta \boldsymbol{e}_j)^+$ remains constant on the set $I$ and this value is the same as $\boldsymbol{x}^+$ on $I$ since $y_a + y_{a+1} + \ldots + y_b = x_a + x_{a+1} + \ldots + x_b$. That is, we have $\boldsymbol{y}^+ = \boldsymbol{x}^+$ in this case.

**Case 2.** Assume that $I \cap J = \emptyset$. As earlier, let $\delta$ be sufficiently small. Write $I = \{a, a+1, \ldots, b\}$ and $J = \{c, c+1, \ldots, d\}$, where $b < c$. Since the isotonic constraint is inactive between the $(a-1)$-th and $a$-th components, the projection $\boldsymbol{x}_I^+$ restricted to $I$ is the same as projecting $\boldsymbol{x}_I$ onto the $|I| = b - a + 1$-dimensional isotonic cone. As $\delta$ is sufficiently small, the projection $(\boldsymbol{x} - \delta \boldsymbol{e}_i + \delta \boldsymbol{e}_j)_I^+$ restricted to $I$ is also the same as projecting $(\boldsymbol{x} - \delta \boldsymbol{e}_i + \delta \boldsymbol{e}_j)_I$ onto the $|I| = b - a + 1$-dimensional isotonic cone.

However, since $i \in I$ but $j \notin J$, we see that $(\boldsymbol{x} - \delta \boldsymbol{e}_i + \delta \boldsymbol{e}_j)_I = \boldsymbol{x}_I - \delta \boldsymbol{e}_i$, where $\boldsymbol{e}_i$ now should be regarded as the $(i - a + 1)$-th canonical-basis vector in the reduced $(b - a + 1)$-dimensional space. Then, by Lemma 1.6 and the following remark, we see that

$$\boldsymbol{y}_I^+ = (\boldsymbol{x}_I - \delta \boldsymbol{e}_i)^+ \le \boldsymbol{x}_I^+$$

in the component-wise sense, which, together with the fact that $y_l^+ = x_l^+$ for $l \in \{1, \ldots, a-1\} \cup \{b+1, \ldots, c-1\} \cup \{d+1, \ldots, n\}$, gives

$$y_1^+ + \cdots + y_l^+ \le x_1^+ + \cdots + x_l^+$$

for all $l = 1, \ldots, c - 1$. Moreover,

$$
\begin{aligned}
y_1^+ + \cdots + y_l^+ - (x_1^+ + \cdots + x_l^+) &= y_a^+ + \cdots + y_b^+ - (x_a^+ + \cdots + x_b^+) \\
&= y_a + \cdots + y_b - (x_a + \cdots + x_b) \\
&= -\delta
\end{aligned} \tag{1}
$$

when $b + 1 \leq l \leq c - 1$.

Now we turn to the case $c \leq l \leq d$. As earlier, for sufficiently small $\delta$, the projection $(\boldsymbol{x} - \delta\boldsymbol{e}_i + \delta\boldsymbol{e}_j)_J^+$ restricted to $J$ is the same as projecting $(\boldsymbol{x} - \delta\boldsymbol{e}_i + \delta\boldsymbol{e}_j)_J$ onto the $|J| = d - c + 1$-dimensional isotonic cone. Then, since $\boldsymbol{y}_J = (\boldsymbol{x} - \delta\boldsymbol{e}_i + \delta\boldsymbol{e}_j)_J = \boldsymbol{x}_J + \delta\boldsymbol{e}_j$, it follows from Lemma 1.6 that

$$
\boldsymbol{y}_J^+ \geq \boldsymbol{x}_J^+, \tag{2}
$$

and meanwhile, we have

$$
y_c^+ + \cdots + y_d^+ - (x_c^+ + \cdots + x_d^+) = y_c + \cdots + y_d - (x_c + \cdots + x_d) = \delta. \tag{3}
$$

Thus, for any $c \leq l \leq d$, (2) and (3) give

$$
y_c^+ + \cdots + y_l^+ - (x_c^+ + \cdots + x_l^+) \leq y_c^+ + \cdots + y_d^+ - (x_c^+ + \cdots + x_d^+) = \delta.
$$

Therefore, we get

$$
\begin{aligned}
&y_1^+ + \cdots + y_l^+ - (x_1^+ + \cdots + x_l^+) \\
&= y_1^+ + \cdots + y_{c-1}^+ - (x_1^+ + \cdots + x_{c-1}^+) + y_c^+ + \cdots + y_l^+ - (x_c^+ + \cdots + x_l^+) \\
&= -\delta + y_c^+ + \cdots + y_l^+ - (x_c^+ + \cdots + x_l^+) \\
&\leq -\delta + \delta \\
&= 0,
\end{aligned}
$$

where the second equality follows from (1).

Taken together, the results above show that

$$
y_1^+ + \cdots + y_l^+ \leq x_1^+ + \cdots + x_l^+
$$

for $1 \leq l \leq d$, with equality when $l \leq a - 1$ or $l = d$. In addition, this inequality remains true—in fact, reduced to equality—when $l > d$. This completes the proof.

$\square$

## 2 Proofs for Section 3

The proof of Theorem 4 relies on the following lemma, which generalizes Lemma 1.1. The proof of this lemma follows the same reasoning as in Lemma 1.1.

**Lemma 2.1.** *Let $\boldsymbol{x} = (x_1, \ldots, x_n) \succeq \boldsymbol{y} = (y_1, \ldots, y_n)$ in the sense that $x_1 \geq y_1, x_1 + x_2 \geq y_1 + y_2, \ldots, x_1 + \cdots + x_{n-1} \geq y_1 + \cdots + y_{n-1}$ and $x_1 + \cdots + x_n = y_1 + \cdots + y_n$. Let $I_1, \ldots, I_m$ be a partition such that $I_1 = \{1, 2, \ldots, n_1\}, I_2 = \{n_1 + 1, \ldots, n_1 + n_2\}, \ldots$. Let $\boldsymbol{x}^+$ and $\boldsymbol{y}^+$ be the projections of $\boldsymbol{x}$ and $\boldsymbol{y}$ onto the isotonic cone $\{\boldsymbol{r} : \boldsymbol{r}_{I_1} \geq \boldsymbol{r}_{I_2} \geq \cdots \geq \boldsymbol{r}_{I_m}\}$, respectively. Then, we have $\boldsymbol{x}^+ \succeq \boldsymbol{y}^+$.*

*Proof of Theorem 4.* Without loss of generality, assume that $R_1 \geq R_2 \geq \cdots \geq R_n$ and therefore the true block ranking satisfies $I_1^\star = \{1, 2, \ldots, n_1\}, I_2^\star = \{n_1 + 1, \ldots, n_1 + n_2\}, \ldots, I_m = \{n_1 + \ldots + n_{m-1} + 1, \ldots, n\}$. For an appropriate permutation we must have $\boldsymbol{y} \succeq \pi \circ \boldsymbol{R} + \boldsymbol{z}$. Thus we finish the proof by invoking Lemma 2.1.

$\square$

*Proof of Theorem 5.* The proof of this theorem follows immediately by noting

$$
\pi_1 \circ \boldsymbol{R} + \boldsymbol{z} \succeq \pi_2 \circ \boldsymbol{R} + \boldsymbol{z}.
$$

$\square$

*Proof of Proposition 3.6.* Without loss of generality, assume $R_1 \geq R_2$. We complete the proof by considering several difference cases regarding fixed $z_1, z_2$, and $\lambda$. First, consider the case where $z_2 > R_1 - R_2 + z_1$ and $\lambda \geq (R_1 + z_2 - R_2 - z_1)/2$. Then, when $(y_1, y_2) = (R_1 + z_1, R_2 + z_2)$, the adjusted scores of reporting the true ranking are $(\frac{R_1 + R_2 + z_1 + z_2}{2}, \frac{R_1 + R_2 + z_1 + z_2}{2})$, and are $(R_1 + z_1, R_2 + z_2)$ if reporting the opposite ranking. When $(y_1, y_2) = (R_1 + z_2, R_2 + z_1)$, the adjusted scores of reporting the true ranking are $(R_1 + z_2, R_2 + z_1)$, and otherwise are $(\frac{R_1 + R_2 + z_1 + z_2}{2}, \frac{R_1 + R_2 + z_1 + z_2}{2})$. As is clear, we have

$$U(\frac{R_1 + R_2 + z_1 + z_2}{2}) + U(\frac{R_1 + R_2 + z_1 + z_2}{2}) + U(R_1 + z_2) + U(R_2 + z_1) \geq +U(R_1 + z_1) + U(R_2 + z_2) + U(\frac{R_1 + R_2 + z_1}{2}$$

since $(R_1 + z_2, R_2 + z_1) \succeq (R_1 + z_1, R_2 + z_2)$. The proof of the remaining cases are similar.

$\square$