# OpenReview forum: "You Are the Best Reviewer of Your Own Papers: An Owner-Assisted Scoring Mechanism"
_NeurIPS.cc/2021/Conference — NeurIPS 2021 Poster_

### Official Review · Reviewer_3cVW · 2021-07-16

**Rating:** 6
**Confidence:** 3

**Summary:**

The author studies the problem of accurate evaluation of papers via peer review. They propose a mechanism by which authors may submit rankings of their submissions ordered by quality, which can then be used to augment scores by solving a convex program.

They show that under a convexity assumption on owners' utility functions, the best strategy from owners is to submit honest rankings of their own items. Subject to this behavior, they then show that the Isotonic Mechanism improves accuracy overall.


**Ethical Concerns:**

None.

**Limitations And Societal Impact:**

Yes.

**Main Review:**

First of all, I recognize the irony of peer reviewing a paper about (improving noisy and bad) peer review. I would like to say that I really think this is an important problem, as reviewing for conferences is a notoriously noisy process, and the author's proposed mechanism is very interesting. Overall, incentivizing authors to truthfully rank their papers (and perhaps other papers...?) could indeed be useful.

I found the paper interesting and and well-presented, but I have some qualms about the (rather strong) assumptions made in the paper.

I found the convexity assumption throughout the paper to be quite strong and perhaps unrealistic. It seems that some authors may indeed reap diminishing returns from more papers being accepted; at least anecdotally, this seems to be a bigger driver than increasing visibility of more papers. I would be interested to see an extension of the theory in the paper to non-convex (perhaps submodular?) functions.

With respect to the distribution of papers per author at conferences, it would also be very enlightening to see a complete distribution of the number of papers per author for large conferences like NeurIPS and ICML. I would hazard a guess that the majority of papers are submitted by authors who only have one paper, and thus there will be a large fraction of papers with no additional information about them. Perhaps this will disadvantage the authors with fewer submissions. One other suggestion that could be very interesting would be to allow authors to evaluate their own papers in comparison with other submissions, but to do this in a way such that their own paper's ranking is not affected by their report, a la notions of impartiality in the ranking literature.

Also, if all authors submit exactly two papers and there is a nice partition of the papers into pairs such that each paper is compared to exactly one other paper, how much does this help increase the accuracy of reviews? It seems like most of the theoretical results really kick in with noisy reviews and many papers per author.

It also seems like the author is perhaps "leaving some money on the table" with respect to multiple owners. If instead of a partition, they allowed for an overlapping collection of subsets where each author with multiple papers was allowed to rank her own papers, this could yield much richer information across subsets that (if everyone reported accurately and honestly) could help the mechanism considerably. Have you considered this option?

Also, I think that this mechanism could be readily abused by reviewing rings. For instance, if an author has multiple submissions and knows that her strongest paper is being reviewed by people in her reviewing ring, then she can report that it is her weakest paper and presumably benefit from the highly positive reviews given to her now-"weakest" paper.


Minor comments:

In Thm 3, it would be nice if you formally defined risk.

224: Theorem 1, Theorem 2
351-2 repeats 348-9

**Time Spent Reviewing:**

2

---

> ### Author Response · Authors · 2021-08-11
> **Response to Reviewer 3cVW**
>
> We appreciate the thoroughness and thoughtfulness of your comments. We are happy to hear that our paper is interesting and well-presented. Below we respond to all the comments raised in your review report in a point-by-point manner.
>
> - **The convexity assumption.** This is indeed a strong assumption and we have acknowledged this fact in the discussion section. It is very true that “some authors may indeed reap diminishing returns from more papers being accepted,” another way to (approximately) put this is to say that the utility function depends on the total number $n$ of submissions. For example, we can let $U_n(r) = \frac{U(r)}{n^{1/3}}$, where $U(r)$ is the utility function when the author submits only one paper. As such, the marginal return diminishes as $n$ increases, and all results in this paper still hold. Nevertheless, we acknowledge that it is not clear how to extend our results when the utility function is not additive (with the exception of Schur-convex function in Section 3). It is unlikely that such nice results (reporting truthfully) would hold in the nonconvex setting, because as Reviewer dEv7 pointed out, *averaging* arising from convexity here plays an essential role. We’ll discuss all these in detail in the revised version, provided our paper would be accepted. We understand that it is not a perfect solution to your important question and hope you can understand.
>
> - **The distribution of papers per author.**
>    - Thanks for this question! As far as I am aware, the following link gives a detailed analysis of ICLR 2020: https://github.com/shaohua0116/ICLR2020-OpenReviewData
> In fact, the issue of “authors submitting a few papers” might not be as serious as it seems. This is because Algorithm 1 seeks to partition the papers according to as few authors as possible. As such, most papers would appear in long rankings, at least in the ideal case.
>
>    - Besides, this will *not* disadvantage the authors with fewer submissions. Instead, this might advantage these authors. This is because in Algorithm 1, these authors are unlikely to be selected (since its $w_o$ is small, so it will not be given by the argmax operation). Then, we’ll use the raw scores for these authors and we can show that the isotonic regression reduces the utility compared with using the raw scores: $U(y_1) + \cdots + U(y_n) \ge U(\widehat R_1) + \cdots + U(\widehat R_n)$ by the Hardy-Littlewood-Polya inequality. This outcome is not a bad thing for the machine learning conference nowadays as it would slightly penalize if authors who made little contribution are included. I’m not sure if I understand how an author can rank her papers in comparison with other submissions. I guess this way the author would always rank her papers higher than the others in her best interest.
>
>    - This is an interesting example. However the isotonic mechanism wouldn’t improve the raw scores much if all authors submit exactly two papers. This is because Theorem 3 says that the improvement is more significant if $n$ is large.
>
> - **Leaving some money on the table.** Thanks for asking this question! In the case of overlapping subsets, different authors may give inconsistent rankings. In particular, consider two authors (A and B) who wrote one very good paper together and they also have several (mediocre) papers written separately. Suppose author A truthfully reports her best paper as the first in the ranking. Then author B has the incentive to rank their common (best) paper as the lowest, thereby increasing the scores of her other papers. This example shows the difficulty arising from combining rankings of overlapping subsets.
>
> - **Reviewing rings.** To remedy this issue, we have to set strict policies to constrain and penalize behaviors that violate academic integrity. For the isotonic mechanism itself, one possibility is to enhance the robustness of this mechanism. For example, we can consider the extension “Soft constraints” in Section 3.
>
> Thanks for the very helpful minor comments and we’ll revise our paper accordingly. Last, we understand that the isotonic mechanism in its present form needs much work before it can be employed. Your comments are very insightful and important, and some are currently beyond the reach of the mechanism. That said, one potential contribution of our work is to attract more researchers to pay attention to this important problem and to improve on the isotonic mechanism. For example, the NeurIPS committee sent to all authors on June 1 to ask them *to rank their papers in terms of their own perception of the papers' scientific contributions to the NeurIPS community*. This gives the isotonic mechanism a firm point to start with.

---

### Official Review · Reviewer_a48h · 2021-07-16

**Rating:** 7
**Confidence:** 3

**Summary:**

This paper introduces the Isotonic Mechanism, which uses the ranking data from authors to get rid of the noisy scores from reviewers. Theoretically, if using the proposed mechanism, the best strategy is to report the correct ranking, which justifies the correctness of the proposed mechanism.

**Limitations And Societal Impact:**

A theoretical paper. No big potential negative societal impact.

**Main Review:**

Overall, the proposed mechanism is interesting and seems very easy to be implemented. The main theorem guarantees that the authors will truthfully report the ranking. The presentation of this paper is also great. The main message is easy to understand, and all theorems are followed with explanations. The problem solved in this paper is indeed critical. Sometimes I also got frustrated by the quality of some reviewers. Theorem 3 is great! It says that the risk can be reduced significantly when $n$ is large.

I have the following three doubts about this paper:
1. The assumption about the utility function might be too strong. For example, if I am the first author for an okay paper and the fourth author for a good paper.  The optimal strategy of mine might be ranking the okay paper on top of the good paper.  Not sure whether the idea of Isotonic Mechanism can also solve this problem if the author ranking is taken into account.
2. I am a bit doubtful about the performance of Algorithm 1 in real-world applications. It might be possible that many owners only have one paper, which makes Isotonic Mechanism reduce to the standard mechanism.
3. I am not very familiar with the related works. Is the author-wise ranking-based approach new in the review system? I think the authors need to provide a clear answer to this question.

**Time Spent Reviewing:**

3 hours

---

> ### Author Response · Authors · 2021-08-11
> **Response to Reviewer a48h**
>
> We appreciate the thoroughness and thoughtfulness of your comments! We were excited to hear that our paper is interesting and the problem solved in this paper is critical. Below we respond to all the comments raised in your review report in a point-by-point manner.
>
> - Thanks for pointing out this important question! To address this question, one possibility is to assume different utility functions for different papers. Another approach is to slightly reduce the ranking length: for example, many large machine learning groups produce a large number of submissions, where the last author is the same person (professor/principal investigator); in this case, most papers should be weighted about the same from the PI’s perspective. We’ll discuss this issue more in the revised version.
>
> - Thanks for this question! As seen from Theorem 3, the isotonic mechanism performs better when the ranking is longer. In the case where many authors have one or two papers, an important fact is that most papers in machine learning have multiple authors. Indeed, the essence of Algorithm 1 is to partition the papers according to a relatively small number of authors such that most papers appear in long rankings.
>
> - To the best of our knowledge, this author-wise ranking-based is new. Perhaps the most relevant work is *J. Wang, I. Stelmakh, Y. Wei, and N. B. Shah. Debiasing evaluations that are biased by evaluations*, which was cited in our submission version. This reference is, however, concerned with the setting where the authors are asked to rate their received reviews.

---

### Official Review · Reviewer_cAEK · 2021-07-16

**Rating:** 6
**Confidence:** 4

**Summary:**

This paper looks at the question improving the process of peer review by allowing authors to submit a ranking of their own papers. The idea is to use this ranking from the authors to improve the noisy rankings coming from the reviewers. The authors show that one can formulate this problem as an isotonic regression problem and that if the authors are utility maxamizers for the sum utility of a set of papers then it is rational to submit and hones ranking of the papers. Extensions are also proposed including authors submitting bucket rankings as well as variants of utility functions.

**Main Review:**

This paper is well written and addresses and important topic, namely how to do more with the available reviewing resources at large conferences. The proposed method is interesting and contains some good ideas for one way to achieve this goal: namely though the use of self assessment.

However I have several issues with the paper: (1) it is not clear that it is within scope for a computing conference as there is no learning and even the optimization algorithms are not detailed in full. (2) The model as it is currently established seems inadequate to deal withe full complexity of the problem. And (3) there is significant related work that is missing, specifically the large literature on impartial peer review under noise models.

I'll address each of these shortcomings in turn.

*Relevance:* The paper proposes an interesting model that incorporates information in a unique and creative way. However, it is unclear to me that this paper fits within the conference call as it does not address learning, optimization, or even computation in a direct way. Optimization algorithms are left to a reference and despite the availability of real world conference data (e.g., PrefLib or the ICLR data) there no empirical or numerical experiments. To be more precise: while it is true that none of these datasets have self reported rankings one could either run purely numerical experiments, e.g., draw from a general distribution such as a Mallows model and then run the suggested algorithm or one could impute the required orders and draw samples from the existing datasets. All the papers I suggested below have experiments with various adaptations such as this and given the bounds I would have liked to see evidence that this proposed method performs well in practice for *some* set of assumptions.

*Model Adequacy:* The model detailed in the paper assumes that authors have a non-decreasing utility function over their own papers. This is a fundamental assumption of the model and the stragegyproofness of the mechanism rests firmly on this assumption. I also find this assumption to be completely inadequate. It seems more reasonable that there are only two outcomes: accept and reject, and I have a 0/1 preference over this. The paper dismisses this model out of hand, which I find strange.

Additionally, the noise model is one of random noise, which seems reasonable, but in the literature it seems much more reasonable to use e.g., Mallows models or some kind of Random Utility Model to model the noise in these rankings. I would have liked to see various noise models discussed or at least cited (see below) since there are several that are widely used.

On line 88 it seems that reviewers have to score all possible papers? This seems completely unfeasible and goes against the arguments above. In all the work in this space (below) we assume some small set of rankings and it's not clear to me how this model would work without the assumption of full utility function elicitation (this is made worse by the lack of computational experiments noted above).

Finally, while it is true that some authors submit many many papers (more than they can possible read) this is being addressed through limits in the number of submissions at many conferences. While you provide stats that many authors provide many papers, it seems that (from the given stats) many more authors have only one paper or perhaps two. I would like to see some discussion of what happens when many of the self rankings are singletons, but this very important case is not discussed.

*Related Work:* I include a few references here but there is significant work in the mechanism design and computational social choice areas that deal with the problem of impartial peer evaluation and review. While the paper current includes references to some work on de-biasing evaluations the much more relevant work on mechanism design (which is what this paper is) is missing. While I think the algorithms in this paper are new, this oversight is significant and worrying.

Aziz, H., Lev, O., Mattei, N., Rosenschein, J.S. and Walsh, T., 2019. Strategyproof peer selection using randomization, partitioning, and apportionment. Artificial Intelligence, 275, pp.295-309.

Noothigattu, R., Shah, N. and Procaccia, A., 2021. Loss functions, axioms, and peer review. Journal of Artificial Intelligence Research, 70, pp.1481-1515.

Jecmen, S., Zhang, H., Liu, R., Shah, N., Conitzer, V. and Fang, F., 2020. Mitigating Manipulation in Peer Review via Randomized Reviewer Assignments. Advances in Neural Information Processing Systems, 33, pp.12533-12545.

Mattei, N., Turrini, P. and Zhydkov, S., PEERNOMINATION: Relaxing Exactness for Increased Accuracy in Peer Selection. IJCAI 2020.

In all this paper contains a nice idea: use self reporting data in a clever way to improve peer review. However, in its current form it does not do enough to show that this method is applicable to the real problem or demonstrate effectiveness of the method. Finally important related work is missing.

-----
Review revision after discussion and rebuttal.

The authors have answered many of my questions on the paper and I have raised my score on this submission. I am now convinced the paper is in scope for the conference but perhaps a few words as to the fit for a machine learning conference are appropriate.

However, I still think that a more comprehensive review of the related literature is necessary for this paper -- there are many works in this general area and situating this method is important. Additionally, after discussion I still feel adding synthetic experiments would greatly improve the paper in that it would give some evidence of the quality of improvement of the bound from Theorem 2 since there there is only a \geq guarantee it would be good to measure this empirically on some imputed data (e.g. a Mallows model). While I agree there will be improvement, some experiments demonstrating this would greatly improve the paper.

In general I still feel that the model for binary outcomes can and should be investigated in future work as it would greatly expand the scope of the paper's application.


**Time Spent Reviewing:**

1.5

---

> ### Author Response · Authors · 2021-08-11
> **Response to Reviewer cAEK**
>
> We appreciate the effort you put in and the comments you provided. However, we must say that we respectfully disagree with some of the comments. Below we respond to all the comments raised in your review report in a point-by-point manner.
>
> - **Relevance.** We fundamentally disagree with this comment. Indeed, our paper fits into the NeurIPS call perfectly.
>
>   - First, ideas and techniques from learning, statistics, and optimization are heavily integrated into the development of this paper. For example, this paper is built on top of isotonic regression, which is a popular technique in statistics and machine learning. Our paper also involves convex optimization, though it is true that this part is a bit indirect from an implementation viewpoint. This is because there are already many optimization methods for solving isotonic regression, and so we didn’t attempt to reinvent the wheel. Nevertheless, it is an interesting optimization problem to solve thousands of isotonic regression problems (Algorithm 1). Due to space constraints, this problem shall be left for future research.
>
>    - Second, it is not feasible to use the public datasets for illustration. This is because Theorem 2 of our paper, which is the main result showing advantages of using the isotonic mechanism, requires to know the underlying true scores, which are not available in these datasets. On the other hand, it is possible to perform experiments on synthetic data. We have used the implementation of isotonic regression in sklearn (https://scikit-learn.org/stable/modules/generated/sklearn.isotonic.IsotonicRegression.html) to perform a quick experiment, showing the improvement brought by our mechanism (Theorem 2). We hope to add these experimental results to the revised version.
>   - Third (but perhaps most important), improving peer review for large machine learning conferences is in urgent need. The potential impact of any improvement is substantial, given the vast applications of machine learning in modern society. Working in machine learning, we wrote this paper because we felt it was our responsibility to recognize this issue and contribute as much as we can. Indeed, soon after the submission of this paper, an email titled *NeurIPS 2021 author survey to better understand expectations of the review process* was sent on June 1 to all authors, asking them *to rank their papers in terms of their own perception of the papers' scientific contributions to the NeurIPS community*. This is clear evidence that our paper is a perfect fit for the NeurIPS call.
>
> - **Model Adequacy.**
>   - In addition to largely determining acceptance or rejection, the review scores also reveal important information (for example, influence on spotlight, oral and best papers). This kind of important **cannot** be modeled by the binary outcome 0/1 alone. Hence, it is reasonable to let the utility function take the review scores as input. Second, our setting only involves review scores and rankings. By contrast, the binary outcome requires additional information (for example, AC’s opinion). In this sense, it is beyond the scope of our paper to directly model the binary outcome.
>   - Thanks for bringing up the noise models. We’ll cite and discuss in the revised version. Nevertheless, this literature is **not** directly related to our paper. This is because the main assumption of our paper is that the author has knowledge of the true ranking, without any noise. That said, it is a very interesting question to extend the isotonic mechanism to stochastic rankings.
>   - The prototype isotonic mechanism requires the author to rank all papers the author wrote. We believe it is feasible. Even granted that it were not feasible, in Section 3 we introduced *Ranking in a block form*, which only requires the author to partially rank the papers he/she wrote.
>    - Thanks for this useful observation! The isotonic mechanism is most ideal when the author has a number of papers (say, 4 or more). Even in the case where many authors have one or two papers, an increasingly true fact is that most papers in machine learning have multiple authors. Motivated by this fact, in Section 2.1 we proposed Algorithm 1. The essence of this algorithm is to partition the papers according to as *few* authors as possible such that most papers appear in long rankings.
>
>
> - **Related Work.** Thanks for suggesting references! We’ll cite and discuss these papers in the revised version. However, after quickly skimming these papers, we’d like to point out that these papers are *not* directly related to the isotonic mechanism. This is because the settings of these papers do not involve the ranking supplied by the author. This entirely distinguishes our paper from the others. Having said this, we’ll add these papers to the references of our paper.

---

### Official Review · Reviewer_dEv7 · 2021-07-16

**Rating:** 8
**Confidence:** 4

**Summary:**

The paper introduces a novel perspective to conference peer review, while the author involves in the review process. For an author with multiple papers, the author rates all her papers. Subsequently, the reviewer scores are projected using isotonic regression, subject to the author ranking. If the author has a convex utility over the papers, the optimal action to report the truthful ranking. The paper also considers many extensions, including multiple paper authors, block form ranking, non-separable utility functions, true-score dependent utility, and using soft constraints instead of isotonic regression.


**Ethical Concerns:**

The current discussion should be sufficient.

**Limitations And Societal Impact:**

The current discussion should be sufficient.

**Main Review:**

The idea in the paper is very interesting and the theoretical analysis is comprehensive. I think it is a very good paper and I would recommend it for acceptance.
Using the author's opinion is something that modern conferences should try.
As is also mentioned in the paper, the current results in the paper clearly still have limitations. Here I add a few points/questions in addition:

1. Extension to multiple authors/owners: In the paper, this is proposed in a separate section and no theoretical results are proved. Why is it better to use larger partitions? Can we use multiple owners' rankings to create better rankings?

2. Multiple reviewers: In practice, the area chairs will look at all scores. The current paper only considers one set of scores y (which can be a summary of the reviews). What if we still want to have multiple sets of scores from different reviewers?

3. The choice of isotonic regression and convex utility function is quite important - I think mostly it is because isotonic regression tends to average scores when the ranking is violated. If an author favors high scores over even scores (convex utility) then she should report truthfully. Other combinations of the author's utility and the mechanism are also great future work.

Other minor points:

1. Line 61: There are double "with the"

2. Line 290: I think "fault tolerance" usually refers to the transformation of input noise (noise of owner ranking) to output noise (the scores R). This section is saying that the owner tends to minimize noise, which is not fault tolerance of the method itself.

3. Line 327: Should be $R_{\rho(i)}$.


----------------
Update: I have read the rebuttal.


**Time Spent Reviewing:**

3

---

> ### Author Response · Authors · 2021-08-11
> **Response to Reviewer dEv7**
>
> We appreciate the thoroughness and thoughtfulness of your comments. We were excited to hear that our paper is interesting and the analysis is comprehensive. Below we respond to all the comments raised in your review report in a point-by-point manner.
>
> - Algorithm 1 seeks to partition the papers according to as few authors as possible. As such, most papers would appear in long rankings. This is because Theorem 3 says the isotonic mechanism would bring more significant benefits when the ranking is longer. It is also challenging to incorporate multiple authors’ rankings due to possible inconsistency between different rankings.
>
> - Thanks for this question! A simple solution is to average over all review scores. We can also incorporate the weights (confidence) into the averaging. We’ll discuss this point in detail in the revised version.
>
> - Thanks for bringing up this direction for future research! The assumption of convex utility is critical, though not necessarily always holds. It is indeed an important problem to extend this assumption.
>
> Thank you very much for spotting three minor points! We’ll revise accordingly in the next version.

---

### Decision · Program_Chairs · 2021-09-27

**Decision:**

Accept (Poster)

**Comment:**

Researchers in machine learning and many other fields have regularly complained about the quality of reviews. This paper proposes a novel idea to mitigate the "noise" in the decisions in a peer-reviewed conference. The idea leverages the known fact from statistics and optimization that given a ranking of papers, isotonic optimization (finding the closest answer under the given ranking) can yield significant reduction in noise. The key (very nice) idea in this paper is to ask authors to provide a ranking of their own submissions. The scores given by the reviewers are then projected on these provided rankings. The paper shows that under certain assumptions, authors are incentivized to report their true perceived rankings of their authored papers, which is then expected to yield the noise-reducing benefits of isotonic optimization.

I have read the paper carefully myself, and the reviewers and I uniformly agree that this is a novel contribution to an important problem. Overall, this paper is a "novel but imperfect" paper. The assumptions here make the current proposal not-yet-practical. Moreover, the paper claims that the assumed conditions are "mild" whereas our assessment is that these assumptions are very strong. (More on this below.) However, I recommend acceptance due to the really fresh perspective offered by this paper to a problem of interest to the NeurIPS community. (I am aware of the typical bias against novel papers in peer review.)

Interestingly, the paper is also timely given the NeurIPS 2021 experiment of authors reporting the rankings of their papers (although NeurIPS 2021 is not using it for making decisions, unlike the proposal of this paper).

With this acceptance, please ensure the following three action items in the camera ready version:

(A) Please include a thorough discussion in the main text regarding the strong assumptions and resulting challenges.  It is important to discuss them not only to ensure that the reader gets an accurate understanding of the work, but also to facilitate follow-up research towards taking this novel idea to practice.

(B) Please incorporate the reviewers' comments and the items promised in the rebuttal.

(C) Please see the comments at the bottom of this meta-review regarding *experiments* quantifying the *magnitude* of benefits, and act accordingly.

Based on the reviews as well as my reading of the paper, here is a summary of some issues that arise with the mechanism due to its strong assumptions.
- - -
1. Convex utilities: This can incentivize reporting of wrong ranking
The paper assumes convex utilities, without any supporting evidence. In my opinion and that of multiple reviewers, the utility for most authors will be quite non-convex with a step-like upward curve near the acceptance threshold. If the utility is non-convex then it incentivizes reporting of the wrong ranking. Here is an example. Suppose I am submitting two papers: one is I think a near-sure accept, and the other is just below bar. Then suppose I rank the worse paper as higher. In this case, under the isotonic method, the top paper's scores will reduce a bit and the worse paper's score will improve a bit. Then my top paper will still get accepted (and hence ranking wrongly didn't really affect me much for this paper) whereas the worse paper has a much higher chance of acceptance under the wrong ranking!

The claim "threshold utilities may not be a sensible choice for modeling the behavior of the user" made in the paper is unsubstantiated and hard to believe. Please remove such claims unless there is supporting evidence.

2. L2 error
It is known that isotonic optimization is good for L2 error. But why is the L2 error a good metric for peer review? In practice, making the accept/reject decision accurately will be considerably important but that is not captured in the L2 error metric analyzed here. Please clarify its purpose in the revision. It is ok to say that this is chosen for theoretical convenience, if that is the case.

3. Does not account for self-selection, and can incentivize submission of low quality papers
Suppose I have a paper that is below the bar (which in my opinion is worse off than all my other submissions) and I am not planning to submit it in the conventional setting (I don't care about this paper since it is not in good shape yet). However if the isotonic method is employed, then the following happens. I rank this paper as worst. If the ratings received are the lowest then it doesn't affect anything. However, if due to noise, it gets rated higher than other papers, then the isotonic method will increase the rating of my other papers, thereby increasing their chance of acceptance. Hence submitting a low quality paper can help me get my other papers accepted.

4. Not accounting for other strategic behavior: This gives undesirably high power to the reviewer
There are a number of issues of dishonest behavior by reviewers in peer review. One example is where a reviewer doesn't like an author or an author's line of research and so deliberately tries to reject their work. This method will give even more power to this dishonest reviewer. For instance, if this reviewer is reviewing that author's work, then giving a low score to that paper will not only reject this paper but under the isotonic mechanism, can also get other papers by this author rejected. As another example, if there is a reviewing ring, if an author has multiple submissions and knows that her strongest paper is being reviewed by people in her reviewing ring, then she can report that it is her weakest paper and presumably benefit from the highly positive reviews given to her now-"weakest" paper.

5. Exchangibility and authors' stress
Submitting papers is already quite stressful for authors, and the dependence of the decisions on these self reports may significantly add to the stress felt by authors. In particular, it can require authors to estimate the review process, and not just their own paper, which can be stressful.  Also, what exactly should you ask authors? To rank papers in order of their perceived chances of acceptance? Or in terms of their scientific contribution?  These two things need not be the same. The assumption on exchangeability of noise is unnatural, for instance, it is known that novel papers suffer from biases against novelty and interdisciplinary papers also suffer in the peer review process.

6. "Guest authorship"
The process of publishing in single blind venues has a problem of guest authorship, where a researcher is made an author to give increased visibility or chances of acceptance to the paper even though that researcher has made no real contribution. Will this mechanism further incentivize guest authorship -- somebody who can provide a ranking that can increase chances of a paper getting accepted?
- - -

The authors should revise the paper to make the shortcomings very clear. For instance, the abstract calls the conditions "mild", which does not represent the nature of assumptions accurately, and was disconcerting. Please clarify the nature of the assumptions here and elsewhere in the paper. The exchangability condition on noise is also called "mild" in the paper, which as discussed above, is not. Please expand the discussion section to discuss these issues in the main text. Section 2.3 can be shortened to make room for this. It is important to discuss them not only to ensure that the reader gets an accurate understanding of the work, but also to facilitate follow-up research to make this novel idea practical.

Minor point: In my opinion, calling it "author-assisted" instead of "owner-assisted"  will be more clear.

Other points raised in the discussions: Reviewers were concerned about the lack of experiments. I had a number of back-and-forth exchanges with the reviewers on this. We discussed that there is no real data available on authors' perceptions of their own papers and no ground truth on papers to validate. As for synthetic experiments, we discussed that there is a good amount of literature showing the magnitude of benefits of isotonic optimization, but the author should either cite literature empirically quantifying the magnitude of benefits of isotonic optimization (not just orderwise but in terms of actual values) or even better include synthetic simulations in the appendix to illustrate this. There was also some confusion where a reviewer misunderstood that the authors have to rank all submitted papers, and this confusion was clarified in a discussion between the AC and reviewer.